# META COMPRESSION:
# LEARNING TO COMPRESS DEEP NEURAL NETWORKS

## ABSTRACT

Deploying large pretrained deep learning models is hindered by the limitations of realistic scenarios such as resource constraints on the user/edge devices. Issues such as selecting the right pretrained model, compression method, and compression level to suit a target application and hardware become especially important. We address these challenges using a novel meta learning framework that can provide high quality recommendations tailored to the specified resource, performance, and efficiency constraints. For scenarios with limited to no access to unseen samples that resemble the distribution used for pretraining, we invoke diffusion models to improve generalization to test data and thereby demonstrate the promise of augmenting meta-learners with generative models. When learning across several state-of-the-art compression algorithms and DNN architectures trained on the CIFAR10 dataset, our top recommendation shows only 1% drop in average accuracy loss compared to the optimal compression method. This is in contrast to 25% average accuracy drop achieved by selecting the single best compression method across all constraints.

## 1 INTRODUCTION

Deep neural networks (DNNs) are being increasingly adopted in diverse domains including computer vision (Redmon et al., 2016), natural language processing (Vaswani et al., 2017), and speech recognition (Amodei et al., 2015). The size of state-of-the-art DNNs has exponentially grown in recent years (Bernstein et al., 2021), allowing them to achieve or even exceed human-level performance for a variety of tasks (Alzubaidi et al., 2021; Silver et al., 2018). Despite this unquestionable benefit, these DNNs have become so large that in some cases they cannot run on a single GPU. Smaller models cannot still fit the limited resources of edge servers or end devices such as mobile phones and smart objects in the Internet of things. This work specifically addresses such a challenge to enable efficient computation, closer to end users.

Several approaches have been proposed to design compact yet efficient DNNs – in terms of both accuracy and performance – by explicitly targeting resource-constrained devices, for instance. On the one hand, model compression takes an existing DNN and applies techniques such as pruning and (or) quantization to obtain a simplified model. The obtained model is smaller, therefore, also faster to execute for inference. This approach has been shown to obtain a significant reduction in model size with close to negligible loss in accuracy. Despite its potential, the performance of the compressed can be only characterized a posteriori: a source DNN has to be first compressed and only then evaluated. As a consequence, tailoring a model to fulfill certain requirements results in a trial-and-error process that depends on appropriately setting the parameters of the specific compression method.

On the other hand, network architecture search explores a design space to find a model that fulfills application-specific requirements in an automated way. This approach effectively enables tailoring the characteristics of a DNN architecture, for instance, based on hardware-specific constraints. However, it entails carrying out an extremely large amount of computation, exceeding several weeks of training. Even worse, the process generally needs to be repeated from scratch for each target configuration. Recently, more advanced techniques have been introduced to train a so-called once-for-all network that can be specialized for diverse target configurations without re-training. However, these schemes significantly constrain the design space and still require substantial computation.

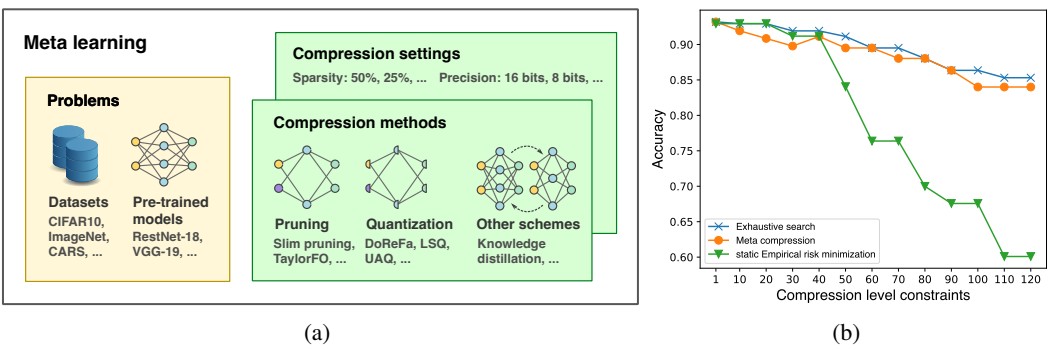

Figure 1: (a) Given several datasets and models pre-trained on them, meta compression predicts the accuracy of a source model when compressed with different methods and settings, *without* running it. (b) Meta compression outperforms a recommendation strategy based on empirical risk minimization for high compression levels (at least 50 times), with a close-to-optimal prediction accuracy.

This article bridges the gap between these two extremes by introducing **meta compression**, a novel approach based on meta learning to simplify a source DNN into one that fulfills application- and device-specific constraints. Our approach aims at answering the following questions: Which is the simplest DNN that achieves a certain accuracy? Which is the most accurate DNN that can run on a given device? It does so by predicting the accuracy of a compressed DNN based on a collection of problems, consisting of several reference DNN models, compression methods, and their settings (Figure 1).

Our approach relies on pre-trained DNN models as a key design choice to reduce the computational overhead of meta learning. Such a choice entails several challenges related to the partitioning of the source source dataset the DNN was trained on. For instance, the training / testing split may not be known, or the data not used for training may be not enough for meta learning. For this reason, we leverage diffusion models to augment the original dataset as a fundamental operation to achieve high prediction accuracy.

We extensively evaluate our proposed framework by using several popular DNNs trained on the CIFAR10 dataset. We obtain 92% top-5 recommendation accuracy compared to 15% top-4 accuracy achievable using a static compression algorithm. Our top recommendation is only 1% far from the optimal compression method in terms of average accuracy loss. Our analysis also reveals that meta compression generalizes well to new datasets, new architectures, and new compression methods.

## 2 RELATED WORK

**Model compression**. DNN compression broadly encompasses techniques aiming at simplifying a source model into a smaller one (Hoefler et al., 2021). Among them, the major approaches are quantization, to reduce the precision of operands, and pruning, to sparsify a network by removing "unimportant" weights or activation values (Gholami et al., 2021; Liang et al., 2021). Both benefit from fine-tuning to restore accuracy (Sanh et al., 2020) and can jointly be applied for further reduction in model size (Hu et al., 2021; Yang et al., 2020; Frantar & Alistarh, 2022). However, characterizing the performance of specific compression methods have been elusive. (Kuzmin et al., 2023) introduce an analytical framework to compare pruning and quantization. However, their results are limited to magnitude pruning and uniform quantization as considered separately. In contrast, our solution allows to predict the accuracy of diverse compression methods, including those leveraging fine-tuning, even when jointly applied.

**Network architecture search**. Network architecture search aims at automatically selecting a model that can satisfy certain properties (Elsken et al., 2019). In this broad context, AutoNBA (Fu et al., 2021) devises a search strategy over networks, bitwidths, and features for hardware acceleration. To reduce computation, OFA (Cai et al., 2019) takes a reference DNN as a backbone to build a so-called once-for-all network. Such a network is trained in such a way that it can be tailored for hardware-specific constraints at a later stage, without re-training – the latter phase achieves pruning as a side effect. The same approach is extended in APQ (Wang et al., 2020) to also consider quantization.

Figure 2: (a) The inner loop of the meta learning process involves training a model to predict the performance of a classifier once compressed with a certain method and related settings. (b) Once learned, the accuracy prediction model can be leveraged to recommend the best compression method for a given problem.

However, APQ only supports channel pruning and uniform quantization. Instead, our approach allows to plug-in arbitrary pruning and quantization schemes as compression methods.

**Meta learning**. Meta learning involves iterating over ML tasks while an outer training loop optimizes meta parameters (Hospedales et al., 2022) and has been applied to different scenarios (Garg & Kalai, 2018; Verma et al., 2020). Metapruning (Liu et al., 2019) applies meta learning to predict weights of a channel-pruned DNN, which is in turn employed to derive a specific channel pruning strategy. Instead, our approach applies meta learning to predict the performance of a compressed model and is not limited to pruning channels. In a different context, a few solutions have recently leveraged diffusion models in the context of meta learning. Among them, Meta-DM (Hu et al., 2023) performs data augmentation to improve the performance of few-shot learning, whereas the work in (Nava et al., 2023) carries out zero-shot task adaptation with both classifier and classifier-free guidance. In contrast, we introduce a framework for meta compression where diffusion models generate validation data to improve prediction performance.

## 3 META COMPRESSION

We now introduce the system model and describe our meta-compression scheme (Figure 1). For clarity, we first address a single problem and compression method, then extend our consideration to a collection of them.

### 3.1 OVERVIEW AND SYSTEM MODEL

A source *dataset* $D = \{T, F, E, V\}$ consists of different, disjoint sets: $T$ for training, $F$ for fine-tuning, $E$ for evaluation, and $V$ for testing. A *classifier*[1] $c$ is pre-trained on $T$. Given these, a *problem* is defined as $p = (c, F, E)$. A *compression method* $K$ uses the pre-trained classifier $c$ and the fine-tuning dataset $F$ to obtain a compressed classifier $c' = K(c, F)$. The loss of the compressed classifier is assessed over the evaluation set $E$ to obtain the true accuracy $A$. An *accuracy prediction model* $g$ takes metadata as input – namely, the features devised by the meta compression – to derive the predicted accuracy $\tilde{A}$.

Accordingly, the training of the prediction model follows the process in Figure 2a. Metadata extraction involves computing problem features, compression method features, and compressed classifier performance (see Section 3.3). A given classifier $c$ compressed by using a configuration $i$ with $K_i$, then the performance of the compressed classifier $c'_i$ is assessed by using the evaluation dataset $E$. In summary, training the meta predictor $g$ involves learning a mapping from problem-specific and compression method-specific features to the compressed classifier performance.

We can proceed to the recommendation once the accuracy prediction model is trained. The first type of recommendation involves selecting the most accurate compression model that satisfies device-specific constraints. To do so, we first prepare a shortlist of compression methods along with their settings. We then refine the candidate options by restricting to those that satisfy the given constraints, for instance, expressed in terms of model size or target compression ratio. The best compression method can be selected by a simple $\arg\max$ function over the predicted accuracy values (Figure 2b).

---

[1] We consider classification problems here for clarity, even though our formulation is general and can be applied to different domains. An application to regression tasks is provided in Section B.4.2.

The second type of recommendation entails selecting the method achieving the highest compression while satisfying an application-specific accuracy target. For this purpose, we consider all available compression methods as predictions and prepare a shortlist of those fulfilling the accuracy constraint. Similar to the previous case, we use the $\arg\max$ function to obtain the method that achieves the highest predicted compression.

It should be now clear how meta compression can answer the research questions introduced earlier. It remains to see how meta learning is effective across multiple problems and compression methods.

## 3.2 PROBLEM STATEMENT

So far we have limited our attention to a single problem $p$ and a given compression method $K$, specialized into $K_i$ according to different configurations. We now extend our consideration to a collection of problems and compression methods, and precisely state the problem we are addressing.

In classification problems, $D$ is a labeled dataset $(\mathcal{X}, \mathcal{Y})$ and $c = L(T)$ is a classifier obtained by learner $L : (\mathcal{X}, \mathcal{Y})^* \to \mathcal{Y}^{\mathcal{X}}$ – the set of all functions from $\mathcal{X}$ to $\mathcal{Y}$ – based on the training set $T \sim \mu^n$. A function $l : \mathcal{Y} \times \mathcal{Y} \to [0, 1]$ is such that $l(y, \hat{y})$ is the loss when the prediction is $\hat{y}$ and the true label is $y$. $l_\mu(c) = \mathbb{E}_{(x,y)\sim\mu} [l(y, c(x))]$ computes the expected loss of a classifier over some data distribution $\mu$.

Now, consider a compression method $K : \mathcal{Y}^{\mathcal{X}} \times (\mathcal{X} \times \mathcal{Y})^* \to \mathcal{Y}^{\mathcal{X}}$ such that $c' = K(c, F)$. Note that we drop the index $i$ from $K_i$ for convenience. The loss of the compressed classifier can be evaluated by using the evaluation dataset $E$. To evaluate the loss of a compression method $K$, consider a distribution over problems $p \sim \nu$. Each problem specifies a pretrained classifier $c \in \mathcal{Y}^{\mathcal{X}}$, and retraining and evaluation dataset $F, E \in (\mathcal{X} \times \mathcal{Y})^*$. Then, the expected loss of a compression method can be defined as

$$l_\nu(K) = \mathbb{E}_{p\sim\nu} [l(y, K(c, F)(x))]$$

Given a family of compression methods $\mathcal{K}$ that satisfy certain compression level constraints, consider the task of selecting the best performing compression method across several problems. A practical objective is to select the compression method with the lowest empirical error across multiple problems, namely, to carry out an empirical risk minimization (ERM). Accordingly, consider a collection $P$ of problems, containing problems $(c_\tau, F_\tau, V_\tau) = p_\tau \in P$. Then,

$$\mathrm{ERM}_{\mathcal{K}}(P) \in \arg\min_{K\in\mathcal{K}} \sum_{p_\tau \in P} \sum_{(x_t,y_t)\in E_\tau} l(y_t, K(c_\tau, F_\tau)(x_t))$$

**Theorem 1.** *For any finite family $\mathcal{K}$ of compression algorithms, any distribution $\nu$ over problems $p_\tau$, and any $n \geq 1$, $\delta > 0$,*

$$\Pr_{P\sim\nu^n} \left[ l_\nu\left(\mathrm{ERM}_{\mathcal{K}}(P)\right) \leq \min_{K\in\mathcal{K}} l_\nu(K) + \sqrt{\frac{2}{n}\log\frac{|\mathcal{K}|}{\delta}} \right] \geq 1 - \delta$$

Solving $\mathrm{ERM}_{\mathcal{K}}(P)$ gives the single best compression method across *all problems* in the collection. By focusing on a specific problem, it is further possible to select the compression method that is best suited to it. Hence, given problem specification $p_\tau$, consider the task of choosing the best compression method among candidates $\{K_1, \ldots, K_m\}$. To solve this, consider problem features $\phi(c_\tau, F_\tau, E_\tau) \in \Phi$, and compression method features $\gamma(K) \in \Gamma$. Also consider a family $\mathcal{F}$ of functions $f : \Phi \times \Gamma^m \to \{1, \ldots, m\}$, which select a compression method among $m$ based on extracted features. Then, the objective of finding the function that selects the best compression method based on ERM can be defined as

$$\arg\min_{f\in\mathcal{F}} \sum_{p_\tau \in P} \sum_{(x_t,y_t)\in E_\tau} l(y_t, K_{f(\phi(p_\tau),\gamma(K_1),\ldots,\gamma(K_m))}(c_\tau, F_\tau)(x_t)) \tag{1}$$

We further simplify the design of recommendation function $f$ by using a loss prediction function $g : \Phi \times \Gamma \to \mathbb{R}_{\geq 0}$, that predicts the loss of a compressed classifier $c'$ for a given problem $p_\tau$ and compression method $K_i$.

$$g(\phi(p_\tau), \gamma(K_i)) \approx l_\mu(K_i(c_\tau, F_\tau))$$

Accordingly, the ERM objective for learning $g$ is

$$\underset{g \in \mathcal{G}}{\arg\min} \sum_{p_\tau \in P} \sum_i \left[ g(\phi(p_\tau), \gamma(K_i)) - l_{\mu^e}(K_i(c_\tau, F_\tau)) \right]^2 \qquad (2)$$

Once $g$ has been learned using sufficient data, $f$ can be described in terms of $g$ as

$$f(\phi(p_\tau), \gamma(K_1), \ldots, \gamma(K_m)) = \underset{i=\{1,\ldots,m\}}{\arg\min} g(\phi(p_\tau), \gamma(K_i)) \qquad (3)$$

This finds the best compression method for any problem $d_\tau$. It naturally follows that we also find the best compression methods across all problems, which was the ERM objective of learning $f$. This formulation of $f$ has an additional benefit, it allows us to solve this performance-constrained compression maximization problem as well using the same accuracy prediction function $g$.

### 3.3 META FEATURES AND ACCURACY PREDICTOR

We use gradient boosted decision trees (Chen & Guestrin, 2016) as the accuracy prediction model. (White et al., 2021) consider numerous models for predicting the performance of DNNs for NAS. Their findings show that the XGBoost model is often the best choice in the low query time regime, which is also our focus.

Obtaining a compressed classifier requires applying a compression method $K$ to a pretrained classifier $c$ and possibly retraining using tuning dataset $F$. Thus, making predictions about the compressed classfier performance requires problem features $\phi(p_\tau)$ and compression method features $\gamma(K_i)$. To accurately evaluate the behavior of the compressed classifier, we also need an evaluation dataset $E$ that is separate from the tuning dataset. We describe these features in more detail in this section.

**Problem features** $(\phi(p_\tau) \in \Phi)$. A problem $p_\tau = (c_\tau, F_\tau, E_\tau)$ consists of a pretrained classifier, and tuning and evaluation dataset. To describe the pretrained classifier, we extract two set of features, architecture features and solution features. Architecture features encode information about the modules used as building blocks of the DNN architecture. Solution features are used to describe the particular solution learned from training the model. These include norms of weights, gradients, and loss and accuracy of $c$ evaluated using $E$.

**Compression method features** $(\gamma(K_i) \in \Gamma)$. The compression process typically consists of iterative pruning and retraining, followed by quantization and retraining. We consider several popular pruning and quantization methods in our experiments. Each pruning method is encoded using a unique pruning identifier, and target sparsity level. Similarly, each quantization method is encoded with a unique quantization identifier and target level.

**Compressed classifier performance**. We evaluate the loss and accuracy of the compressed classifier on evaluation dataset $E$ to be used as the ground truth for predictions.

The accuracy prediction model $g$ takes in problem features and compression method features, and predicts the compressed classifier performance. The complete process of learning the accuracy prediction model $g$, and obtaining compression method recommendations using it is described in the next section.

## 4 EXPERIMENTAL EVALUATION

We conduct extensive experiments to evaluate the efficacy of the proposed meta compression algorithm against the state of the art in pruning and quantization (Liu et al., 2017; Esser et al., 2020). We also study the impact of two design choices: choice of evaluation data and choice of meta features. Finally, we evaluate the generalization performance of our meta learner to new evaluation data, new architectures, and new compression methods. The performance of the recommended compression method is always evaluated using the test set $V$, which is left untouched until this step.

While we conduct an extensive analysis for classifiers trained for CIFAR10 (Krizhevsky, 2009) classification task in this section, we provide additional results in the appendix to help gauge generalization to larger dataset problems, using an ImageNet (Deng et al., 2009) experiment in B.4.1, and

generalization to regression tasks using models provided in the tabular dataset benchmark (Gorishniy et al., 2021) in B.4.2. Also note that while the following evaluation is focussed on the compression constrained accuracy maximization task, it proposed should works for the accuracy constrained compression maximization task as well, for which we present some results in B.4. The code for reproducing the experiments is available at *https://anonymous.4open.science/r/DeeperCompression-4EED/*

## 4.1 EXPERIMENTAL SETUP

### 4.1.1 ARCHITECTURES

Our setup consists of several popular DNN models trained on the CIFAR10 dataset consisting of VGG19 (Simonyan & Zisserman, 2014), ResNet18 (He et al., 2015b), GoogLeNet (Szegedy et al., 2014), DenseNet121 (Huang et al., 2016), ResNeXt29-2x64d (Xie et al., 2016), MobileNet (Howard et al., 2017), MobileNetV2 (Sandler et al., 2018), DPN92 (Chen et al., 2017), SENet18 (Hu et al., 2017), ShuffleNetV2 (Ma et al., 2018), RegNetX-200MF (Radosavovic et al., 2020), SimpleDLA (Yu et al., 2017), with their implementations in (Kuang-Liu, 2017). Unless stated otherwise, $F$ consists of 20% randomly sampled images from the CIFAR10 training dataset, $E$ consists of 10k images generated using a diffusion model trained on CIFAR10 train dataset, and $V$ is comprised of the complete CIFAR10 test dataset. Metadata extraction for a single pruning,quantization level (40 retraining epochs) takes around 40 minutes on a Tesla A100 GPU. This data is extracted for 10 different sparsity and 4 different quantization levels, totaling 40 combinations. The total time required to run this configuration is approximately 26 hours (12 GPUs for 1.0833 days, i.e., 13 GPU days) considering parallel execution on 12 Tesla A100 GPUs for 12 different architectures.

### 4.1.2 COMPRESSION METHODS

**Pruning**. We consider different families and types of pruning methods. Specifically, we consider the following for unstructured pruning: *magnitude pruning*, consisting of a simple weight pruning scheme; and *pruning+tuning*, as the former followed by retraining for 40 epochs. For structured pruning, instead we consider: *L1 norm pruning* (Li et al., 2016), which removes weights with low L1 norms; *network slimming* (Liu et al., 2017), which masks scaling factors in later batch normalization layers to prune channels in convolution layers; and *TaylorFO* (Molchanov et al., 2019), which prunes convolutional layers based on the first-order Taylor expansion on weights.

**Quantization**. We consider the following quantization methods: *uniform affine quantizer* (Krishnamoorthi, 2018), which maps continuous values to discrete levels by scaling and rounding within a given quantization range; *Dorefa-net* (Zhou et al., 2016), which represents the weights and activations through a ternary code to save storage, in addition to employing a mix of fixed-point quantization, scaling, and rounding to reduce quantization error; *learned step-size quantization* (Esser et al., 2020), which employs a learnable scaling factor and a fixed quantization step size.

## 4.2 RECOMMENDATION PERFORMANCE

We evaluate the recommendation performance of the proposed meta compression algorithm by considering whether any of the top-k recommendations for a given constraint performs at most $\epsilon$ worse than the optimal recommendation. We compute top-k recommendation accuracy by averaging this across several constraints. We also compute top-1 error, which reports how far off is the top recommendation from the optimal recommendation in terms of compressed model accuracy (on average). For comparison, we also consider selecting the single best pruning and quantization method on a dataset of problems and recommending the same for all architectures and compression level constraints. We find that Slim pruning (Liu et al., 2017) and learned step size quantization (Esser et al., 2020) perform the best based on empirical risk minimization. The top-1 static strategy involves quantizing to 4 bits with LSQ and we adapt Slim pruning rate to match the compression level provided in the constraint. For this reason, we consider the top-4 metric[2] for the fixed strategy due to the four quantization levels – 32, 16, 8, and 4 bits – and matching pruning levels to achieve desired compression.

---

[2] For a fair comparison, we report the same top-4 accuracy for meta compression where applicable.

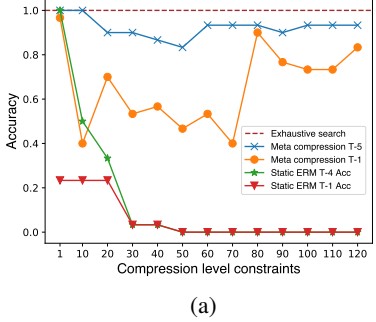

| Metric | Meta Compression | Static ERM |
|---|---|---|
| T5 Accuracy | **0.92** | 0.15 |
| T1 Accuracy | **0.66** | 0.06 |
| T1 Error | **0.01** | 0.25 |
| MAE | **0.10** | 0.12 |
| Kendall Tau | **0.65** | - |

(a)                                                                      (b)

Figure 3: (a) Comparison of dynamic recommendation of meta compression against static recommendation of optimal ERM based on the state of the art, namely, the combined used of Slim pruning and LSQ (Liu et al., 2017; Esser et al., 2020). (b) Recommendation performance of prediction.

| Compression Algorithm | T5 Acc. | T1 Acc. | T1 Error | MAE |
|---|---|---|---|---|
| Prune + Quant | 0.97 | 0.92 | 0.02 | 0.14 |
| Level-prune + Dorefa-quant | 0.97 | 0.65 | 0.04 | 0.07 |
| L1-prune + Dorefa-quant | 0.93 | 0.45 | 0.04 | 0.08 |
| TaylorFO-prune + Dorefa-quant | 0.85 | 0.38 | 0.06 | 0.08 |
| Slim-prune + Dorefa-quant | 0.99 | 0.70 | 0.03 | 0.18 |
| Level-prune + LSQ-quant | 0.98 | 0.78 | 0.01 | 0.02 |
| L1-prune + LSQ-quant | 0.99 | 0.64 | 0.02 | 0.06 |
| TaylorFO-prune + LSQ-quant | 1.0 | 0.64 | 0.03 | 0.07 |
| Slim-prune + LSQ-quant | 0.98 | 0.72 | 0.02 | 0.17 |

Table 1: MAE of the predictor $g$ trained for different compression algorithms.

We start by splitting the set of architectures into train set and test set. The meta prediction model is trained for architectures in the train split, and used to predict performance of architectures in the test split. We perform several such splits to remove any bias towards a specific split. We set tolerable accuracy drop $\epsilon$ to 0.01 and vary the minimum compression constraint between 1x to 120x, with a step size of 10. For each constraint, we compare the predictions made using our meta prediction model against the optimal choice found using exhaustive search. The average results across multiple splits are reported in Figure 3a. The figure clearly shows how the proposed Meta Compression approach outperforms fixed recommendation with ERM. Figure 3b reports aggregate metrics across all compression constraints, and we see that the top-5 performance is 92% and the average error is only 0.01 compared to 0.25 for static recommendation with ERM. In other words, one of our top-5 recommendations performs close to the optimal compression algorithm 92% of times and the performance of the recommended compression method is off by only 1% accuracy on average. The Kendall Tau correlation of our predictions is also comparable to the results reported in (White et al., 2021) despite having a more complex learning task, as it involves predicting the performance of new architectures further modified by applying different compression methods.

The recommendation performance of different compression methods is detailed in Table 1. It provides a breakdown of how well our recommendations work for different compression methods, for the same setup as that used for Figure 3. The purpose of the table is: to understand whether certain compression methods are significantly easier to predict; and to compare retraining-based compression methods against one-shot compression methods. Interestingly, the results show that mean absolute error (MAE) in accuracy prediction for one-shot pruning and quantization (Purne+Quant) is on average worse than the same for compression algorithms with retraining. This could be explained by the fact that retraining shifts and concentrates the compressed model accuracies towards higher accuracy values. However, the top-5 recommendation performance is still high, suggesting that the meta predictor is often able to order input configurations correctly even when the accuracy predictions for specific input configurations are worse on average.

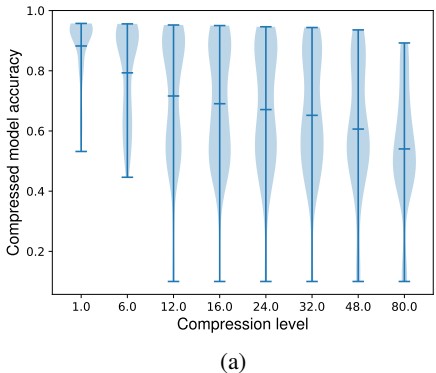 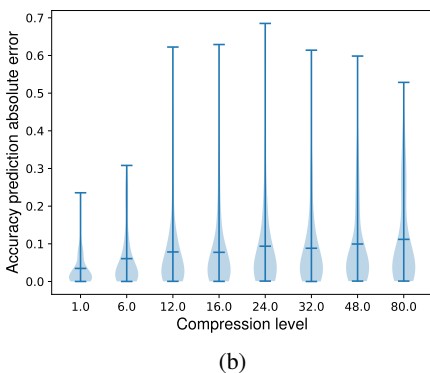

(a)                                                     (b)

Figure 4: (a) Compressed model accuracy and (b) absolute error in accuracy prediction for different compression levels

Figure 4 illustrates the distribution of compressed model accuracies at various compression levels grouped into equal sized bins for different metrics. Specifically, Figure 4a shows the prediction performance of learned prediction model $g$ for the samples in those bins. We see that the prediction error is slightly better at very low compression levels, where the compressed models are also more accurate. The average prediction error stays roughly the same with increasing compression levels, however, the variance of error does grow. Further increasing the compression levels beyond those in the figure considerably reduces accuracy (below 50%), making the sheer value of accuracy not meaningful – for these reason they were not reported. Figure 4b shows the absolute error in prediction accuracy for different compression levels. The results highlight that such an error is consistently low, even for compression levels of 48x and 80x. The variance is more pronounced in the higher range of the absolute error, although most of the values are concentrated below the mean.

### 4.3 DESIGN CONSIDERATIONS

Applying the proposed meta learning to predict compressed classifier performance involves making several design considerations. One of them is the choice of the prediction model $g$. An empirical analysis has revealed that feed forward DNNs did not provide improvement over gradient boosted decision trees (Chen & Guestrin, 2016), which were chosen as the prediction model $g$. Other key choices involve selecting evaluation data and the meta features. Analysis of various meta features can be found in the appendix. Considerations involved in selecting evaluation data are discussed next.

#### 4.3.1 EVALUATION DATA

The primary metric under consideration when choosing evaluation data $E$ is generalization performance to test data. For satisfactory outcomes, it is desirable that the evaluation data contains samples not seen during the pretraining or fine-tuning phase. In the *ideal* scenario, this could be achieved by hiding some of the train data during the training phase and use it as evaluation data. However, this is often impossible, as the available pre-trained models already make use of the full training data. One could instead sample evaluation data from test data to avoid retraining the models from scratch with partial train data. Unfortunately, this approach has its own drawbacks. First, we risk leaking information from test data, possibly compromising the integrity of the final test evaluation. Second, sampling from test data implies a very limited dataset size. Instead, we propose using diffusion models to generate evaluation data. Such an approach overcomes the limitations mentioned above with remarkable performance.

Table 5b details the impact of data selection on prediction accuracy and MAE. Each row of the table refers to drawing eval data from different sources: the test set, data generated from the entire source dataset with Diffusion-Based Generative Models (Karras et al., 2022), the *ideal* scenario, and the training set. We can see that diffusion model achieves even better top-5 accuracy than the ideal scenario, as also evident from Figure 5a which shows it as a function of the compression level. This

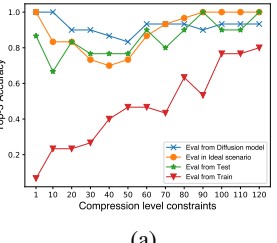

| Eval selection | T5 Acc. | T1 Acc. | T1 Error | MAE |
|---|---|---|---|---|
| From test data | 0.85 | 0.55 | 0.06 | 0.12 |
| **With diffusion model** | **0.92** | **0.66** | **0.01** | **0.10** |
| Ideal | 0.89 | 0.58 | 0.03 | 0.12 |
| From training data | 0.48 | 0.14 | 0.13 | 0.24 |

(a)            (b)

Figure 5: (a) Evaluation data selection choices and (b) Prediction performance for different data selection strategies

could be explained by reduced test performance of pretrained classifiers in the ideal scenario due to reduction in size of the train dataset.

## 4.4 GENERALIZATION PERFORMANCE

We train the meta-predictor across multiple problems and compression algorithms. Therefore, it is extremely important to analyze how well the meta-predictor generalizes to new samples consisting of unseen data, DNN architectures, and compression algorithms. Such an analysis allows to assess how learnable characteristics effectively transfer to unseen samples. For this reason, we carried out several experiments which are described in detail in the appendix. Here we present a summary of our findings.

**New architectures.** The first question is how well a meta-predictor trained on one set of architectures translates to new architectures. On average, the accuracy of our predicted compression method shows only 1% drop in accuracy compared to the optimal compression method, and a top 5 recommendation accuracy of 92%. This suggests strong generalization to new architectures.

**New data.** This option considers what happens if the evaluation data used for training the meta prediction model is not available at test time. When generalizing to new data and new architectures, there is a 2% drop in accuracy of the predicted compression method compared to the optimal compression method. While this is slightly worse than the previous case, the top 5 recommendation accuracy is still 91%. These results also suggest strong generalization to new data.

**New compression methods.** For this setting, we obtain top-5 accuracy of 86% and 11% average drop in accuracy for the predicted compression method. This drop in performance was to be expected, as our compression method features are not descriptive enough to capture similarities and dissimilarities between different compression methods. There is good scope for improvement here by designing more descriptive compression method features.

## 5 CONCLUSION

Machine learning applications rely on deep neural networks (DNNs) that have become substantially complex and large. Training them requires a significant amount of resources and time, leading to energy wastage. These DNNs cannot either be deployed as such at the edge or onto resource-constrained devices. Large, overparametrized DNNs can be effectively slimmed down though. How to achieve a good compression-accuracy tradeoff is a big challenge. We address it by learning how to compress DNNs with a novel meta-learning approach – the first, to the best of our knowledge. We train a regression model to predict the accuracy of a pre-trained DNN after being compressed with a combination of techniques (including pruning and quantization) without having to evaluate it. We take a flexible yet rigorous approach to achieve provably good meta learning. We also carry out an extensive evaluation against state-of-the art compression schemes. The obtained results demonstrate that meta compression is effective, and that diffusion models allow its applications even to scenarios in which evaluation data is scarce, thereby extending its applicability in practice.

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

## A  PROOF OF THEOREM 1

**Theorem 1.** For any finite family $\mathcal{K}$ of compression algorithms, any distribution $\nu$ over problems $d_\tau$, and any $n \geq 1, \delta > 0$,

$$\Pr_{P \sim \nu^n}\left[l_\nu(\mathrm{ERM}_\mathcal{K}(P)) \leq \min_{K \in \mathcal{K}} l_\nu(K) + \sqrt{\frac{2}{n}\log\frac{|\mathcal{K}|}{\delta}}\right] \geq 1 - \delta$$

*Proof.* Let

$$\epsilon = \sqrt{\frac{2}{n}\log\frac{|\mathcal{K}|}{\delta}}.$$

Consider an empirical estimate of $l_\nu(K)$ using some dataset of problems $P \sim \nu^n$, $(c_\tau, F, E) = p_\tau \in P$, given by

$$l_P(K) = \frac{1}{n}\sum_{p_\tau \in P}\sum_{(x_t, y_t) \in E} l(y_t, K(c_\tau, F_\tau)(x_t)).$$

Denote the optimal compression method $K_0 \in \arg\min_{K \in \mathcal{K}} l_\nu(K)$. Writing the Chernoff bound for $l_P(K_0)$, we have

$$\Pr_{P \sim \nu^n}[l_P(K_0) \geq l_\nu(K_0) + \epsilon/2] \leq e^{-2n(\epsilon/2)^2}.$$

Now, consider the set $B$ of bad compression methods that are more than $\epsilon$ far away from $l_\nu(K_0)$. More formally, $B = \{K \in \mathcal{K} \mid l_\nu(K) \geq l_\nu(K_0) + \epsilon\}$. By writing the Chernoff bound for each $K \in B$, we have

$$\Pr_{P \sim \nu^n}[l_P(K) \leq l_\nu(K) - \epsilon/2] \leq e^{-2n(\epsilon/2)^2}.$$

Clearly, $l_\nu(\mathrm{ERM}_\mathcal{K}) \geq l_\nu(K_0) + \epsilon$ holds only if $l_D(K) \leq l_D(K_0)$ for some $K \in B$. Specifically, either $l_P(K) \leq l_\nu(K) - \epsilon/2$ for some $K \in B$, or $l_D(K_0) \geq l_\nu(K_0) + \epsilon/2$. By applying the union bound, the result holds with probability at most $|\mathcal{K}|e^{-2n(\epsilon/2)^2} = \delta$.

## B  ADDITIONAL EXPERIMENTS

### B.1  EXPERIMENTAL SETUP

**ImageNet setup.** We considered the following architectures: VGG11 (Simonyan & Zisserman, 2015), Squeezenet (Iandola et al., 2016), Densenet121 (Huang et al., 2016), Alexnet (Krizhevsky et al., 2012), Resnet (He et al., 2015a) and Shufflenet (Zhang et al., 2017) $F$ consists of 30k labelled images taken from the Imagenet train dataset, $V$ (30k images) and $E$ (20k images) consists of images sampled from the Imagenet validation dataset (50k images) in the ratio of 3:2 respectively.

**XGBoost meta prediction.** The XGBoost model is configured with 100 estimators (number of trees) and a maximum depth of 10 for each tree. The features are preprocessed as follows before being fed to the XGBoost model:

*Categorical Feature Encoding.* The raw data from the dataframe was converted into a format that could be effectively used by the XGBoost model. To achieve this, we encoded categorical features – including dataset identifier, compression method identifier – through a one-hot encoding.

| Metric | Meta Compression | Static ERM |
|--------|------------------|------------|
| T1 Acc | **0.77** | 0.25 |
| T1 Error | **0.08** | 0.44 |
| MAE | **0.12** | 0.13 |

Table 2: Recommendation performance for the ImageNet setup.

*Numerical Features.* Numerical features including loss and accuracy of the pretrained classifier, gradient norms were incorporated after scaling them between 0 and 1. The number of architecture parameters was divided into 10 bins and linearly mapped onto the [0,1] interval.

*Additional features.* Features encoding the 5 largest eigenvalues of Hessian were considered in early experiments, but were dropped after observing low feature importance in predictions as reported in the Appendix B.3.2. Similarly, detailed architecture features were also considered in experiments by encoding each layer into two variables, namely, the layer type (e.g., convolutional, fully-connected) and the layer size.

## B.2    COMPUTE COST DETAILS

**Meta Training cost.** Extracting metadata for a single pruning and quantization level (40 retraining epochs) on ImageNet takes approximately 300 minutes using an AMD MI250x GPU. This process is performed for 10 different sparsity levels and 4 different quantization levels, resulting in a total of 40 combinations. To complete this configuration, it would require around 200 hours in total, utilizing 6 AMD MI250x GPUs running in parallel for 8 days (equivalent to 48 GPU days). This parallel execution accounts for 6 different architectures.

**Meta Recommendation cost.** For CIFAR 10 setup, given a new architecture and compression constraint, it takes around 0.002 seconds to run inference (Meta recommendation) using the XGBoost model on a CPU. Considering a given constraint and a new architecture, there are approximately 36 combinations to explore. Each combination consists of one sparsity level paired with one of four quantization levels (32, 16, 8, and 4 bits). For each compression method (total 9 of them are used) and sparsity/quantization level, the search process takes around 5 minutes. To perform an exhaustive search and identify the optimal compression regime, it is necessary to evaluate all 36 combinations (4 sparsity / quantization combinations for 9 compression method combinations). This search would require approximately 3 hours to complete (36 combinations / 5 minutes per combination) for a single architecture under the given constraint. Similarly for Imagenet setup, given a new architecture and compression constraint, it takes around 0.002 seconds to run inference (Meta recommendation) using the XGBoost model on a CPU. In this case, there are approximately 18 combinations to explore. Each combination consists of one sparsity level paired with one of two quantization levels (32 and 16 bits). For each compression method (total 9 of them are used) and sparsity/quantization level, the search process takes around 20 minutes. To perform an exhaustive search and identify the optimal compression regime, it is necessary to evaluate all 18 combinations (2 sparsity / quantization combinations for 9 compression method combinations). This search would require approximately 6 hours to complete (18 combinations for 20 minutes per combination) for a single architecture under the given constraint.

## B.3    RESULTS

## B.4    ACCURACY CONSTRAINED COMPRESSION MAXIMIZATION

This section presents the results for the accuracy-constrained compression maximization problem, instead of the compression-constrained accuracy maximization results already shown in the paper. The results are shown in Figures 6a and 6b. They show that the predictor has an even better performance in this task, particularly, it achieves a top-5 recommendation accuracy of 98% (as opposed to 92%).

### B.4.1    IMAGENET EXPERIMENT

**Recommendation performance.** We conducted ImageNet recommendation experiments for a smaller set of compression levels due to significantly higher compute cost of fine tuning ImageNet

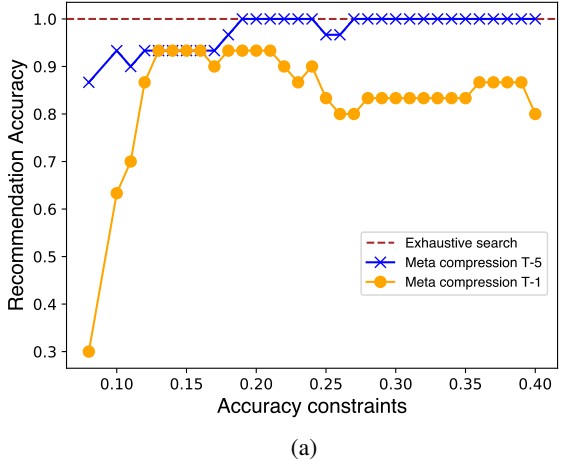

(a)

| Metric | Meta Compression |
|--------|-----------------|
| T5 Acc | **0.98** |
| T1 Acc | **0.84** |
| T1 Error | **0.1** |
| MAE | **0.07** |

(b)

Figure 6: (a) Comparison of dynamic recommendation of meta compression against static recommendation of optimal ERM based on the state of the art, namely, the combined used of Slim pruning and LSQ (Liu et al., 2017; Esser et al., 2020). (b) Recommendation performance of prediction.

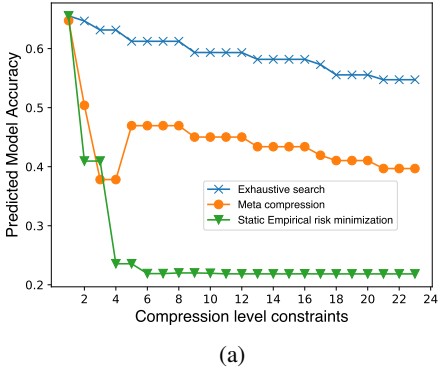

(a)

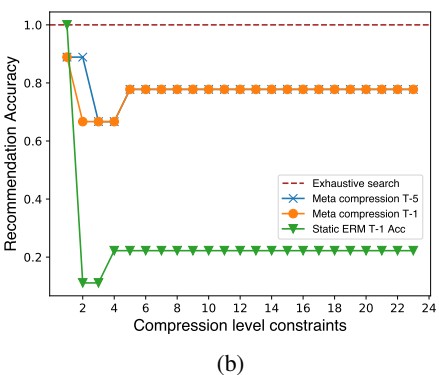

(b)

Figure 7: (a) Average accuracy of models compressed using the top recommended compression method across several compression level constraints. (b) Top-k recommendation accuracy across several compression level constraints..

| Compression Algorithm | T5 Acc. | T1 Acc. | T1 Error | MAE |
|----------------------|---------|---------|----------|-----|
| Prune + Quant | 0.98 | 0.80 | 0.03 | 0.13 |
| Level-prune + Dorefa-quant | 0.69 | 0.49 | 0.16 | 0.25 |
| L1-prune + Dorefa-quant | 0.81 | 0.55 | 0.03 | 0.08 |
| TaylorFO-prune + Dorefa-quant | 0.73 | 0.66 | 0.02 | 0.08 |
| Slim-prune + Dorefa-quant | 0.90 | 0.79 | 0.04 | 0.19 |
| Level-prune + LSQ-quant | 1.0 | 1.0 | 0.01 | 0.10 |
| L1-prune + LSQ-quant | 0.95 | 0.79 | 0.02 | 0.06 |
| TaylorFO-prune + LSQ-quant | 0.97 | 0.75 | 0.02 | 0.06 |
| Slim-prune + LSQ-quant | 0.93 | 0.85 | 0.01 | 0.13 |

Table 3: MAE of the predictor $g$ trained for different compression algorithms for the ImageNet setup.

models. The maximum sparsity level is set to 0.94, quantization precisions considered are 32 bits and 16 bits, and the maximum compression level is set to 24. We have considered 10 different sparsity levels ranging from 0 to 0.94. We evaluate the recommendation performance of the proposed meta compression algorithm by considering whether any of the top-k recommendations for a given

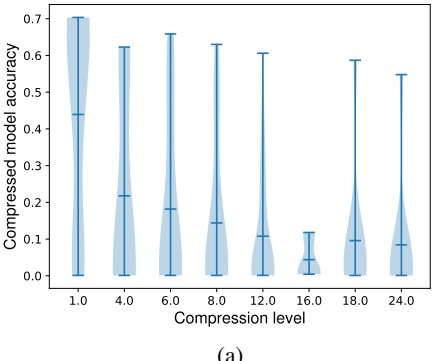 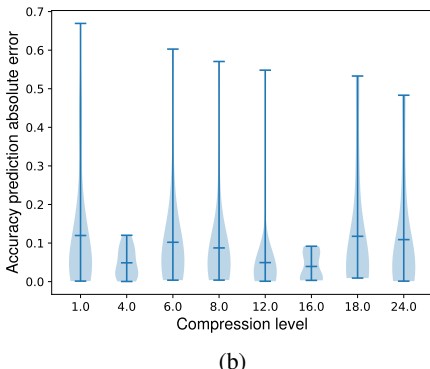

(a)                                         (b)

Figure 8: (a) Compressed model accuracy and (b) absolute error in accuracy prediction for different compression levels for the ImageNet setup

| Eval data selection | T5 Acc. | T1 Acc. | T1 Error | MAE |
|---|---|---|---|---|
| From test data | 0.78 | 0.77 | 0.08 | 0.12 |
| From training data | 0.35 | 0.14 | 0.30 | 0.17 |

Table 4: Recommendation performance for different data selection strategies with the ImageNet setup.

constraint performs at most $\epsilon$ worse than the optimal recommendation. We find that the best static strategy for this case involves quantizing to 32 bits with LSQ and adapting Slim pruning rate to match the compression level provided in the constraint. We set $\epsilon$ to 0.01, same as the CIFAR10 setup, and vary the minimum compression constraint between 1x to 24x, with a step size of 1.

The top-1 recommendation performance and MAE of learned accuracy prediction function $g$ are tabulated in Table 2. We see that the top-1 error reveals 8% average drop in accuracy compared to 44% drop using the static ERM approach. The same observation can also be made from Figure 7, where Meta Compression significantly outperforms the static recommendation strategy at higher compression level constraints.

Table 3 shows the recommendation performance for different compression methods, and Figure 8 shows the distribution of compressed model accuracies and absolute error in accuracy prediction at several compression levels. Compared to the CIFAR10 setup, the accuracy of compressed models drops rapidly as compression level increases. This can be improved with more fine-tuning using more data and retraining epochs, however, also requiring more compute resources. Once again, we can see that the variance of compressed model accuracies is correlated with mean absolute error in accuracy prediction at different compression level ranges. The absolute error values are concentrated below the mean, and the mean always stays below 0.12.

**Eval data selection.** For the ImageNet setup, we conducted experiments with two data selection choices, a) $E$ sampled from train data, and b) $E$ sampled from test data. Table 4 tabulates the results. The behavior is consistent with the observation made for the CIFAR setup, that sampling from test data (and using a reduced testset for final evaluation) performs significantly better than sampling from train data as the feature evaluations done during the meta-training phase generalize well to testing/recommendation phase.

### B.4.2 CALIFORNIA HOUSING REGRESSION EXPERIMENT

We conducted an additional experiment using DNNs trained on the California housing regression dataset. In a similar setup to the one used for obtaining Figure 3a in the paper, we set up an experiment with 4 benchmark architectures from (Gorishniy et al., 2021) (DCNv2, SNN, ResNet, MLP; 3:1 train:test split), 12 different compression levels, and compressed classifier accuracy prediction changed to compressed regressor RMSE prediction. Separate data splits were used for training, evaluation, and testing. RMSE values were clipped to fall in the interval [0,1] (RMSE of all pretrained models is less than 0.51). The obtained results are reported in the table below.

| Metric | T1 Acc. | T1 Error | MAE | Kendall Tau |
|---|---|---|---|---|
| Meta Compression | 1.00 | 0.00 | 0.13 | 0.51 |

Table 5: Meta recommendation and meta prediction performance for the California housing regression experiment.

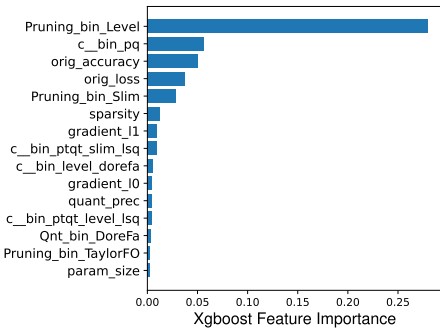

Figure 9: Feature importance for the compressed model accuracy prediction task for combined CIFAR10 and Imagenet setup

Meta compression achieves 100% top-1 recommendation accuracy when predicting the regression performance of a new architecture, clearly demonstrating the applicability of Meta Compression to the tabular dataset domain.

### B.4.3 META FEATURE SELECTION

The learned accuracy prediction model $g$ also offers valuable insights about the specific meta features that contribute the most towards accurate compressed model accuracy predictions. Specifically, decision trees allow us to compute the feature importance by using the Breiman equation (Hastie et al., 2009), which are reported in Figure 9. The results show that a few compression algorithm tags, followed by the loss and accuracy of the pretrained classifier are the strongest predictor which also makes intuitive sense. Target sparsity level is a stronger predictor than target quantization level, which could be due to more sparsity levels and fewer quantization levels being present in the dataset. Interestingly, aggregate gradient metrics such as $L_0$ and $L_1$ norm of gradients have more importance than the number of parameters in the pretrained model. This adds weight to the intuition that observing the slope of the solution learned after pretraining reveals insights about the compression performance of the model. We also experimented with using features such as more descriptive architecture features, and using largest eigenvalues of Hessian to extract second order derivative information, but observed no significant improvement.

### B.5 GENERALIZATION PERFORMANCE

In this section, we describe the methodology used for performing the generalization experiments. The obtained results for the CIFAR10 and ImageNet setup are presented in Figure 10.

**New architectures** (N-Y-N). This is the default setup. It assumes using the same evaluation data and compression methods that have been used during meta training to be also used at the recommendation phase. This resembles a typical deployment scenario where we apply previously known compression methods to new pretrained models. To evaluate this, we split the set of 12 pretrained DNNs for CIFAR10 in a 3:1 train-test ratio, and the 6 models for ImageNet in a 2:1 train-test ration. We report the average the performance across multiple random splits. Worse results for the ImageNet setup can be explained by 2 factors, a) use of $E$ sampled from test set instead of using diffusion model, and b) lack of sufficient training data due to worse train-test split, fewer architectures, fewer compression levels. When comparing the performance when eval data is sampled from test data, we obtain 8% top-1 error for the ImageNet setup which is only slightly worse than the 6% top-1 error obtained for the CIFAR10 setup.

| New data | New arch. | New Compr. | T5 Acc. | T1 Acc. | T1 Error | MAE |
|---|---|---|---|---|---|---|
| No | Yes | No | 0.92 | 0.66 | 0.01 | 0.10 |
| Yes | Yes | No | 0.91 | 0.66 | 0.02 | 0.11 |
| No | No | Yes | 0.86 | 0.34 | 0.11 | 0.13 |
| Yes | No | Yes | 0.85 | 0.34 | 0.12 | 0.15 |

(a)

| New data | New arch. | New Compr. | T5 Acc. | T1 Acc. | T1 Error | MAE |
|---|---|---|---|---|---|---|
| No | Yes | No | 0.78 | 0.77 | 0.08 | 0.12 |
| Yes | Yes | No | 0.78 | 0.77 | 0.09 | 0.13 |
| No | No | Yes | 0.74 | 0.51 | 0.14 | 0.14 |
| Yes | No | Yes | 0.74 | 0.51 | 0.15 | 0.14 |

(b)

Figure 10: Generalization performance of $g$ to new data, architectures, and compression methods for (a) CIFAR10 setup and (b) ImageNet setup.

**New data** (Y-Y-N). This considers the case when the evaluation data used for meta-feature computation during training is not available at the testing phase. We are still giving recommendations for new architectures as in the previous case. The obtained results for both the CIFAR10 setup and the ImageNet setup show only slight drop in performance in this case.

**New compression methods** (N-N-Y, Y-N-Y). This considers generalization to new compression methods while giving recommendations for pretrained models already seen during the meta-training phase. To evaluate this, we split the set of compression specifications into train set and test set based on the compression method used, and keep the same set of pretrained classifiers in both the train set and the test set. For both setups, Figure 10 reveals considerable drop in top-1 error and accuracy but marginal drop in top-5 accuracy compared to the case of new architectures.

