# OpenReview forum: "Meta Compression: Learning to compress Deep Neural Networks"
_ICLR.cc/2024/Conference — Submitted to ICLR 2024_

### Official Review · Reviewer_a7if · 2023-10-23

**Soundness:** 2 fair
**Presentation:** 3 good
**Contribution:** 1 poor
**Rating:** 3
**Confidence:** 5

**Summary:**

The paper addresses the challenges of deploying large pretrained deep learning models in resource-constrained scenarios by introducing a novel meta-learning framework. This framework offers high-quality recommendations for selecting pretrained models, compression methods, and compression levels tailored to specific resource, performance, and efficiency constraints. The proposed approach improves generalization to test data by using diffusion models for limited access to unseen samples.

**Strengths:**

+ The proposed approach formulates the recommendation for compression methods as an optimization problem, finding the solution via boosted decision trees. This framework can somewhat give a solution when we deal with new data that are not accessible.
+ The proposed meta-learning framework shows good performance in experiments.
+ This paper provides theoretical proof of the proposed meta learning optimization.

**Weaknesses:**

- The novelty of this paper is limited. The proposed method is a simple machine learning problem, which is easy to prove in a certain hypothesis.
- In the actual scenarios, the performance of a certain compression method is highly related to multiple hyper-parameters. This paper does not consider these factors.
- The pruning and quantization are usually implemented serially in practice. However, the proposed method considers them separately and then merges them together using a simple criterion.
- The quantization and pruning methods tested in this paper are out-of-date. Also, other compression categories like low-rank matrix and tensor decomposition are not considered.
- According to Fig. 7, the proposed method does not work well on large-scale datasets.
- This paper does not compare with SOTA compression methods (published after 2021) that are directly used to compress models on unseen datasets.
- The training cost is still very high, i.e., 8 days. I don't think the proposed framework can save time a lot as compared to existing SOTA compression methods.

**Questions:**

This paper uses heuristic information based on existing data. How can you guarantee the existing heuristics work well on another dataset as compared to direct compression and tune on that dataset?

---

> ### Author Response · Authors · 2023-11-22
>
> We thank the reviewer for their valuable feedback. Our responses follow.
>
> “_The novelty of this paper is limited. The proposed method is a simple machine learning problem, which is easy to prove in a certain hypothesis._”
>
> We address a fundamental question in this paper - can we predict the compression accuracy tradeoff of a novel pre-trained model without actually compressing the model? We formulate this problem and devise a framework based on meta-learning that solves this problem, building upon strong theoretical foundations. Furthermore, we demonstrate that diffusion models can be leveraged to generate additional evaluation data resembling test distribution in a meta-learning context.
>
> “_In the actual scenarios, the performance of a certain compression method is highly related to multiple hyper-parameters. This paper does not consider these factors._”
>
> We consider two key hyperparameters for compression methods - target sparsity level and target quantization level. Fine-tuning hyperparameters such as learning rate, number of epochs, etc. were kept fixed and decided based on the values used for pretraining.
>
> “_The pruning and quantization are usually implemented serially in practice. However, the proposed method considers them separately and then merges them together using a simple criterion._”
>
> All our experiments involve applying pruning followed by quantization serially. There is no merging criterion. For instance, if the optimal strategy is to use only pruning to 90% sparsity, the meta predictor will suggest pruning to 90% sparsity and quantizing to 32bit precision, which is equivalent to no quantization. In addition, we would like to highlight that the proposed framework can easily be extended to any joint pruning+quantization specification, for e.g. Slim pruning + LSQ quantization, as already discussed in the paper.
>
> “_The quantization and pruning methods tested in this paper are out-of-date. Also, other compression categories like low-rank matrix and tensor decomposition are not considered._”
>
> The choice of compression methods selected in our experiments is based on a practical constraint - the availability of stable implementations that can compress a wide array of DNN architectures. Keeping this in mind, we considered all compression methods that could reliably compress all considered architectures with available implementations in the popular open-source model compression library - Neural Network Intelligence (NNI). We will consider additional compression methods in future work.
>
>
> “_According to Fig. 7, the proposed method does not work well on large-scale datasets._”
>
> We agree that the performance gap between meta compression and exhaustive search is slightly higher in the case of ImageNet setup experiment, but this is mainly because fewer data points were used in the ImageNet setup experiment due to limited available compute in academic settings. However, we would like to highlight here that even in this case, the performance is significantly better than any static recommendation strategy using a single pruning and quantization method across all constraints.
>
> “_The training cost is still very high, i.e., 8 days. I don't think the proposed framework can save time a lot as compared to existing SOTA compression methods._”
>
> While the cost of training the meta-prediction model is comparatively high, the main benefit of using the proposed approach is that obtaining recommendations for a new deployment scenario, consisting of previously unseen pretrained architectures and compression level constraints, is extremely cheap (order of a few milliseconds). Following this, the model can be compressed using the recommended compression method specification. Meta compression outperforms network architecture search approaches here in terms of compute as the latter require significantly more compute for every new deployment scenario.

---

### Official Review · Reviewer_KXmL · 2023-11-01

**Soundness:** 2 fair
**Presentation:** 1 poor
**Contribution:** 2 fair
**Rating:** 5
**Confidence:** 4

**Summary:**

In scenarios where it is vital to make precise choices regarding pretrained models, compression techniques, and compression levels to meet the requirements of a particular application and hardware constraints, this paper introduces a novel approach. The approach utilizes a compression performance predictor to address these challenges, providing customized, high-quality recommendations aligned with specified resource, performance, and efficiency constraints. Additionally, the method incorporates diffusion models to enhance the model's generalization capabilities to test data. To validate the proposed approach, extensive experimentation is conducted, covering a range of state-of-the-art compression algorithms and Deep Neural Network (DNN) architectures trained on the CIFAR-10 dataset.

**Strengths:**

1. The overall concept is intriguing and holds the potential to evolve into a strong research paper. The aim to develop a compression accuracy predictor that offers compression method recommendations for users is noteworthy. However, the current format lacks effective organization and effective illustration.

2. The research is substantiated with evaluations on a diverse range of architectures, and a large training dataset is created for the compression accuracy predictor.

**Weaknesses:**

1. It's advisable to include citations in the introduction to enhance the document's comprehensibility. The absence of citations, except in the initial paragraph, can pose challenges for readers in understanding the content. Incorporating citation references in the introduction will provide valuable context and background information.

2. I find that the overall presentation of the work can be somewhat challenging to follow. For instance, in the description of the inner loop of the meta-learning process in Figure 2a, there appears to be a discrepancy where Metadata extraction lacks information from the compressed classifier, which contradicts the description in the main context.

3. The title suggests that the focus of the work is on "learning to compress DNN" implying the main objective is to compress various models with different compression methods using a meta-learner. However, the paper is more like "learning to choose compression methods of DNN" and predominantly centers on training a compression accuracy predictor to predict performance. It then recommends specific compression methods based on different requirements. This deviation from the title's apparent focus may appear inconsistent or confusing to readers.

**Questions:**

1. The construction process involves numerous architectural considerations and requires training the compressed model as well. While the training cost remains manageable for smaller datasets like CIFAR-10, it's essential to investigate whether this approach introduces excessive overhead when applied to larger datasets such as ImageNet-1000.

2. Could you please offer further elaboration on the "META FEATURES"? Additional details about the specific features used would be beneficial for a clearer understanding.

---

> ### Author Response · Authors · 2023-11-22
>
> We thank the reviewer for their valuable suggestions. We will take them into account for improving the introduction and the illustration in Figure 2a. Please find below the answers to your questions.
>
> “_The construction process involves numerous architectural considerations and requires training the compressed model as well. While the training cost remains manageable for smaller datasets like CIFAR-10, it's essential to investigate whether this approach introduces excessive overhead when applied to larger datasets such as ImageNet-1000._”
>
> While one key benefit of the proposed framework remains that the computation cost of obtaining compression method recommendations for a new deployment scenario is very small (order of a few milliseconds), the scalability of the training time of the meta-learning model is also important, as suggested by the reviewer. The training time of the meta-learning model scales linearly with the time required for 1 epoch training of the base architectures. We have provided results for the ImageNet-1k dataset in the appendix, which were obtained by training the meta-learning model on fewer data points.
>
> “_Could you please offer further elaboration on the "META FEATURES"? Additional details about the specific features used would be beneficial for a clearer understanding._”
>
> Additional details about meta-features and their preprocessing have been provided in Appendix B.1.

---

### Official Review · Reviewer_FBBW · 2023-11-01

**Soundness:** 2 fair
**Presentation:** 3 good
**Contribution:** 2 fair
**Rating:** 5
**Confidence:** 3

**Summary:**

The authors introduce a method utilizing meta learning framework to provide compression methods for any deep learning methods tailoring it for specific resource, performance and efficiency constraints. There main focus is to apply the compression techniques to address the challenges of deploying large models in resource-constrained scenarios.

**Strengths:**

The authors provide a clear motivation of the problem and is easy to read and follow. The authors also supplement the claims with some proofs and empirical results.

**Weaknesses:**

One part which is not very clear is around the choice of the accuracy predictor and not very clear how the experiments look like with respect to other design choices around different compression strategies which would make it better to understand the generalizability of the solution. Because of this choice it limits the applicability of the results as they are compared against only one compression constraint.

It is not clear how the research generalized with more recent modeling techniques specifically around transformers, which would help to strengthen the paper and understand the generalizability of the provided approach.

**Questions:**

1. How would the result and analysis hold against newer architectures specifically around transformers?
2. Do you see limitations of the approach with respect to datasets or architectural choice?

---

> ### Author Response · Authors · 2023-11-22
>
> We thank the reviewer for their valuable feedback. Our responses follow.
>
> _Choice of accuracy predictor_
>
> Two architectures were considered for the accuracy prediction task - multi layer perceptron (MLP) and gradient boosted decision trees (GBDT), with hyperparameters such as depth of MLP and number of decision trees selected using cross validation. GBDT outperformed MLP in our experiments, which is also consistent with the findings of [1] where multiple performance prediction architectures for DNNs are considered.
> [1] White et al, “How Powerful are Performance Predictors in Neural Architecture Search?”, NeurIPS’21
>
> _Choice of compression stragies and generalizability_
>
> The choice of compression methods selected in our experiments is based on a practical constraint - the availability of stable implementations that can compress a wide array of DNN architectures. We found the open source library “neural network intelligence (NNI)” to align the closest with our requirements and used the compression methods implemented there for our experiments. Some methods such as movement pruning had to be discarded as their fine tuning diverged for several architectures leading to unreliable data. In summary, our experiments so far suggest that the proposed meta compression approach works well across all compression algorithms that can reliably compress a variety of DNN architectures.
>
> “_Applicability of the results as they are compared against only one compression constraint._”
>
> While we evaluate and provide results for two key constraints in our analysis - compression constraint and accuracy constraint, additional constraints can also be easily incorporated into the proposed framework. For instance, a constraint on FLOPs/MACs can be implemented by limiting the search space to compression methods that reduce FLOPs(eg. Channel pruning), and predicting the performance of several pretrained models after compression using channel pruning methods to a level that satisfies the constraint.
>
> “_It is not clear how the research generalized with more recent modeling techniques specifically around transformers, which would help to strengthen the paper and understand the generalizability of the provided approach._”
>
> We fully agree that using more recent models and compression methods would be beneficial. However, many of these (including transformers) require significant computing resources and are tailored for specific architectures, making the evaluation very challenging. We will consider such an evaluation as future work.

---

> ### Comment · Reviewer_FBBW · 2023-11-22
>
> Thanks a lot to the authors for providing detailed answers and clarification to the questions asked. I still feel that the baseline comparison against more recent architectures plus at least another compression strategy evaluation would help to strengthen the light on the generalization of the approach, in the current format I would keep my initial assessment.

---

### Official Review · Reviewer_aDqZ · 2023-11-04

**Soundness:** 2 fair
**Presentation:** 2 fair
**Contribution:** 2 fair
**Rating:** 3
**Confidence:** 3

**Summary:**

This paper presents a meta learning algorithm which can predict a good compression algorithm for a given compression target.

**Strengths:**

The idea of using a generative model to generate test data seems like a great solution to the limited data problem. For real-world applications, I think this problem is much less important because new data can always be obtained.

**Weaknesses:**

1) It was hard to contextualize this work for me. For example, how does this work compare to [1], which uses multi-objective Bayesian optimization to yield an entire Pareto-frontier for architecture + compression algorithm hyperparameters?
2) What is the novelty of this work? Is neither a new meta learning algorithm nor compression algorithm, as far as I can tell. To me, it seems the main novelty is the application of meta learning to model compression. In this case, I would need to see experimental evidence that meta learning for model compression is better than existing approaches using Bayesian optimization [1], DNAS, OFA

[1] https://proceedings.neurips.cc/paper_files/paper/2019/file/044a23cadb567653eb51d4eb40acaa88-Paper.pdf

[2] https://proceedings.neurips.cc/paper_files/paper/2022/file/753d9584b57ba01a10482f1ea7734a89-Paper-Conference.pdf

[3] https://hanlab.mit.edu/projects/ofa

**Questions:**

1) I was confused what exactly is the static erm strategy?

---

> ### Author Response · Authors · 2023-11-22
>
> Thank you for your valuable comments; our responses follow.
>
> “_It was hard to contextualize this work for me. For example, how does this work compare to [1], which uses multi-objective Bayesian optimization to yield an entire Pareto-frontier for architecture + compression algorithm hyperparameters?_”
>
> Thank you for suggesting additional references. [1] is closer to network architecture search (NAS) methods cited in our paper, where Bayesian optimization is used to navigate the search space of multiple architectures and pruning levels, in contrast to reinforcement learning which is commonly used in NAS papers. It also has the same key drawback as other NAS methods - that several search iterations are needed for any new deployment scenario that corresponds to a new accuracy, compression level constraint. Note that each step of this search involves training a DNN architecture and evaluating its performance which is quite expensive. We take a different approach which involves learning from multiple data points across DNN architectures, compression methods, and compression levels, which requires more compute in the beginning to generate these data points, but makes the compression method recommendation for any new deployment scenario very cheap (few milliseconds).
>
> “_What is the novelty of this work? Is neither a new meta learning algorithm nor compression algorithm, as far as I can tell. To me, it seems the main novelty is the application of meta learning to model compression. In this case, I would need to see experimental evidence that meta learning for model compression is better than existing approaches using Bayesian optimization [1], DNAS, OFA_”
>
> As correctly pointed out by the reviewer, the novelty of our approach lies in devising a framework based on meta-learning to provide recommendations tailored for specific accuracy-resource constraints. Another novel insight is the successful demonstration of the feasibility of using diffusion models to generate additional data that resembles test distribution in a meta-learning context. We agree with the reviewer that providing additional results for comparison against the stated references will be useful, and we will consider it in the future.
>
>
> “_I was confused what exactly is the static erm strategy?_”
>
> The static ERM strategy involves selecting the single best pruning and quantization method combination that yields the highest performance on average across several accuracy/compression level constraints.

---

### Meta-Review · Area_Chair_114d · 2023-12-02

**Metareview:**

The authors propose a meta learning framework to recommend high quality compression strategies of DNN models under given resource, performance and efficiency constraints.  It is based on a prediction model that predicts the accuracy a DNN model after compression. The authors give a theoretical analysis of the proposed framework and carry out experiments on CIFAR10 to verify its effectiveness.  The topic under investigation is important to the machine learning community but there are concerns raised by the reviewers.  Most of the reviewers consider the novelty of the work is not overwhelmingly significant given various existing similar techniques already in the community such as multi-objective Bayesian optimization.  The authors need to make a comparison with these techniques.  The exposition also needs improvement including the organization and some technical details.  Lastly,  experiments are only conducted on CIFAR10 which is a small dataset in the main body of the paper.  Results on larger dataset (ImageNet) are somehow put in the Appendix.  Given the current form, this paper can not be accepted.  I would suggest the authors make improvements according to the comments by the reviewers and submit it to another venue.

**Justification For Why Not Higher Score:**

Novelty of the work is not significant.  Exposition needs improvement.  Experiments are somewhat limited (only on CIFAR10).

**Justification For Why Not Lower Score:**

N/A

---

### Decision · Program_Chairs · 2024-01-16

Reject